# HLA Genotypes in Patients with Infection Caused by Different Strains of SARS-CoV-2

**DOI:** 10.3390/ijerph192114024

**Published:** 2022-10-28

**Authors:** Ludmila Bubnova, Irina Pavlova, Maria Terentieva, Tatiana Glazanova, Elena Belyaeva, Sergei Sidorkevich, Nataliya Bashketova, Irina Chkhingeria, Mal’vina Kozhemyakina, Daniil Azarov, Raisa Kuznetsova, Edward S. Ramsay, Anna Gladkikh, Alena Sharova, Vladimir Dedkov, Areg Totolian

**Affiliations:** 1Russian Research Institute of Hematology and Transfusion Science, FMBA, 191024 St. Petersburg, Russia; 2Department of immunology, Faculty of medicine, Pavlov First Saint Petersburg State Medical University, Russian Ministry of Health, 197022 St. Petersburg, Russia; 3Saint Petersburg Office, Federal Service for Consumer Rights Protection and Human Welfare, 191025 St. Petersburg, Russia; 4Saint Petersburg Center for Hygiene and Epidemiology, 191023 St. Petersburg, Russia; 5Saint Petersburg Pasteur Research Institute of Epidemiology and Microbiology, 197101 St. Petersburg, Russia; 6Martsinovsky Institute of Medical Parasitology, Tropical and Vector Borne Diseases, Sechenov First Moscow State Medical University, 119435 Moscow, Russia

**Keywords:** COVID-19, HLA, northwestern federal district, Russia, variants of concern

## Abstract

The aggressive infectious nature of SARS-CoV-2, its rapid spread, and the emergence of mutations necessitate investigation of factors contributing to differences in SARS-CoV-2 susceptibility and severity. The role of genetic variations in the human HLA continues to be studied in various populations in terms of both its effect on morbidity and clinical manifestation of illness. The study included 484 COVID-19 convalescents (northwest Russia residents of St. Petersburg). Cases in which the responsible strain was determined were divided in two subgroups: group 1 (*n* = 231) had illness caused by genovariants unrelated to variant of concern (VOC) strains; and group 2 (*n* = 80) had illness caused by the delta (B.1.617.2) VOC; and a control group (*n* = 1456). DNA typing (HLA-A, B, DRB1) was performed at the basic resolution level. HLA-A*02 was associated with protection against infection caused by non-VOC SARS-CoV-2 genetic variants only but not against infection caused by delta strains. HLA-A*03 was associated with protection against infection caused by delta strains; and allele groups associated with infection by delta strains were HLA-A*30, B*49, and B*57. Thus, in northwest Russia, HLA-A*02 was associated with protection against infection caused by non-VOC SARS-CoV-2 genetic variants but not against delta viral strains. HLA-A*03 was associated with a reduced risk of infection by delta SARS-CoV-2 strains. HLA-A*30, HLA-B*49, and HLA-B*57 allele groups were predisposing factors for infection by delta (B.1.617.2) strains.

## 1. Introduction

This work is a continuation of a study that published its results in 2021 [1]. Extensive polymorphism of the human leukocyte antigen (HLA) system is understood to be the result of a need to maintain immunological diversity in the species as a form of protection in the face of various existential threats, including pandemics. The key role that HLA molecules play in the immune response to pathogens, such as foreign antigen presentation, are well known. Human populations also reflect huge molecular variability in HLA alleles. These facts have become the basis for numerous studies aimed at clarifying the influence of HLA genotypes on characteristics of the individual response to infection caused by the severe acute respiratory syndrome-related β-coronavirus 2 (SARS-CoV-2 β) [2,3]. A number of studies have focused on finding specific alleles for susceptibility or resistance to this infection. Individual patterns were shown, but they may represent different relationships depending on level [4,5,6]. An overall pattern may be universal for several populations. A different situation may exist specific to subpopulations since HLA allele distributions vary in different population subgroups. Any subpopulation’s existence depends on many factors, including pathogen presence, immune responses permitting longevity, and host genetic factors (such as HLA type). In this regard, it is clear that analysis of the relationship between individual HLA genotypes and occurrence of SARS-CoV-2 infection, or its clinical severity, should be performed, taking into account the distribution of HLA genes among residents in the study’s corresponding region.

The ongoing coronavirus disease 2019 (COVID-19) pandemic poses new challenges for the scientific community. The emergence of SARS-CoV-2 coronavirus mutations has been regularly observed globally, with some potentially increasing transmissibility and/or virulence: alpha (B.1.1.7); beta (B.1.351); gamma (P.1); and delta (B.1.617.2) [7]. The aggressively infectious nature of SARS-CoV-2, and its rapid spread, necessitate the investigation of factors contributing to observed differences in susceptibility and severity among individuals. Although the delta lineage was not yet present in the human population during the first year of the pandemic, it is critical to continually examine the relationship between emerging SARS-CoV-2 variants and the adaptive immune system. Hamelin et al. found that the most frequent D614G mutation in the S protein was associated with a 15-fold decrease in binding affinity of the mutated epitope for HLA-A*02:01 compared to the reference/non-mutated epitope [8].

The role of genetic variations in the human major histocompatibility complex continues to be studied in various populations in terms of both its effect on morbidity and clinical manifestation of illness. The aim of this study was to compare the distributions of HLA-A*, HLA-B*, and HLA-DRB1* allele groups in residents of Russia’s northwestern region with illness caused by various SARS-CoV-2 strains relative to a control group consisting of people living in the same region.

## 2. Subjects, Materials, and Methods

Subjects. The study included 484 patients (residents of St. Petersburg, Russia) with a novel coronavirus infection (COVID-19) who were potential donors of hematopoietic stem cells (HSCs). They included 246 men (50.8%) and 238 women (49.2%), aged 21 to 66 years. The general group of patients was divided into three subgroups. Group 1 (231 patients) included those with illness caused by viral genetic variants not classified as so-called variants of concern (VOC). Group 2 (80 patients) included those with illness caused by the delta (B.1.617.2) VOC. A third group (173 patients) included cases in which the genetic variant of the virus was not established; these patients were excluded from further subgroup analysis.

Since SARS-CoV-2 genotyping was not performed at the time of illness, the inclusion criterion was date of illness. We also relied on surveillance data for viral genetic variants circulating in St. Petersburg in 2020, which showed several trends. During routine study of SARS-CoV-2 genetic diversity from January 2021 until January 2022, 846 nasopharyngeal swabs from patients with COVID-19, admitted to hospitals or health centers located in Saint Petersburg, were collected and delivered to the Saint Petersburg Pasteur Institute for analysis as part of a related study. All samples of sufficient quality underwent sequencing and phylogenetic analysis [9]. Until the beginning of March 2021, non-VOC genetic variants were detected in the vast majority of cases. Starting from June, the SARS-CoV-2 genetic landscape changed completely. Up to the end of 2021, the delta (B.1.617.2) genetic variant, a VOC, was detected in the vast majority of cases (Figure 1).

Patient group 1 featured 116 men (50.2%) and 115 women (49.8%), aged 21 to 66 years. Patient group 2 featured 44 men (55.0%) and 36 women (45.0%), aged 22 to 53 years. Illness was confirmed by registration in the St. Petersburg System for Automated Registration of Infectious and Parasitic Diseases with the receipt of an epidemiological number (‘SysAutReg Infection’, registration No. 2008615797, dated 4 December 2008). The control group consisted of 1456 volunteers, aged 20 to 60 years, composed of 871 men (59.8%) and 585 women (40.2%). Differences in sex and age between subject groups were insignificant. All persons included in the study were residents of Russia’s northwestern federal district (St. Petersburg, Russia).

Materials. Peripheral venous blood obtained by venipuncture was used for the study. Genomic DNA was isolated from peripheral blood nucleated cells using the 208 010 PROTRANS DNA Box 500 reagent kit (Protrans) according to the manufacturer’s protocol. Assessment of isolated DNA quality and quantity was carried out using spectrophotometry with the Smartspec Plus instrument; quality was assessed by optical density ratio (260/280 nm). The final concentrations of isolated DNA samples were 25–35 ng/µL, with optical density ratios in the range 1.6–1.8.

Methods. Immunogenetic examination (HLA typing) was carried out in accordance with the European Federation of Immunogenetics (EFI) international standards using molecular genetic typing methods. HLA typing was performed at the basic resolution level. Allele groups of HLA genes (loci A, B, DRB1) were determined using: polymerase chain reaction (PCR) with sequence-specific primers (SSP) approved for clinical use produced by Protrans (Hockenheim, Germany); and PCR with sequence-specific oligonucleotide probes (SSOP) manufactured by BAG Health Care (Lich, Germany). PCR with endpoint analysis was performed in a model T100 96-well thermal cycler (Bio-Rad Laboratories, Hercules, CA, USA). For PCR product visualization, the samples were separated using horizontal electrophoresis (2% agarose gel), stained with ethidium bromide, and transilluminated under UV light (320 nm). HLA typing results were determined from primer mix tables updated according to the actual version of the IPD-IMGT/HLA allele database (https://www.ebi.ac.uk/ipd/imgt/hla/ URL (accessed on 4 August 2022).

The SSOP method, using sequence-specific oligonucleotide probes fixed to the bottom of the typing well, was performed in a fully automatic mode using the Mr. SPOT Processor for human tissue genotyping with accessories (BAG Diagnostics GmbH, Lich, Germany). Reaction results were processed and interpreted using HISTO MATCH software.

Both HLA typing methods used made it possible to identify: 21 allele groups of the HLA-A locus (*01, *02, *03, *11, *23, *24, *25, *26, *29, *30, *31, * 32, *33, *34, *36, *43, *66, *68, *69, *74, *80); 36 allele groups of the HLA-B locus (*07, *08, *13, *14, *15, *18, *27, *35, *37, *38, *39, *40, *41, *42, *44, *45, *46, *47, *48, *49, *50, *51, *52, *53, *54, *55, *56, *57, *58, *59, *67, *73, *78, *81, *82, *83); and 13 allele groups of the HLA-DRB1 locus (*01, *03, *04, *07, *08, *09, *10, *11, *12, *13, *14, *15, *16).

Statistical processing of results was performed using population genetics methods and the programs: Microsatellite Tools for Excel. The significance of differences in distributions of HLA allele groups were determined by the Fisher Exact test. Differences were regarded as significant at *p* < 0.05.

## 3. Results

Among all examined, including those in group 1, men and women were equally divided. In group 2 patients (illness caused by the B.1.617.2 delta strain), men slightly prevailed: 55.0% men and 45.0% women. The HLA-A*, HLA-B*, and HLA-DRB1* allele group frequencies among the overall group of COVID-19 convalescents, compared with controls, are presented in Table 1, Table 2 and Table 3, respectively.

As seen from the data presented in Table 1, HLA-A*02 frequency was significantly lower among patients in the overall convalescent group (484 people). In the overall group of convalescents and in the control group, the following HLA allele groups were not detected: HLA-A*34, HLA-A*43, HLA-A*74, or HLA-A*80.

When studying the distribution of HLA-B alleles in both groups, HLA-B*78, HLA-B*82, and HLA-B*83 were not detected.

As seen from Table 2 and Table 3, no specific patterns were noted regarding HLA-B* and HLA-DRB1* allele groups distributions among all examined convalescents in comparison with the control group. An exception was HLA-DRB1*10, which relates to the low frequency of these alleles in the population and the fact that this allele group was not found in the overall convalescent group.

Since the overall convalescent group featured clearly identified patients with illness caused by non-VOC strains (group 1, 231 people), as well as those with illness caused by the delta (B.1.617.2) VOC (group 2, 80 people), immunogenetic features (HLA distributions) were studied by subgroup (1, 2).

The HLA-A*, HLA-B*, and HLA-DRB1* allele groups distributions among individuals who had COVID-19 (groups 1, 2), in comparison with the control group and between these groups, are presented in Table 4, Table 5 and Table 6, respectively. In those who had COVID-19, a slightly lower number of alleles was detected for each of the studied HLA loci than in the control group: 15 for HLA-A* (control 17); 22 for HLA-B* (control 33); and 12 for HLA-DRB1* (control 13). This is likely due to the fact that the control group was substantially larger than the convalescent groups.

As seen from the data presented in Table 4, HLA-A*02 allele group frequency was significantly lower among patients in group 1 (illness caused by non-VOC genetic variants): 42.42% vs. 51.72% in the control group (*p* = 0.0088). This suggests that the HLA-A*02 allele group is associated with a reduced risk of developing illness caused by non-VOCs.

Among patients with illness caused by the delta (VOC) genetic variant, HLA-A*02 allele group frequency did not differ from that in the control group. Furthermore, in this patient group, HLA-A*30 frequency was higher by more than 2-fold (8.75% versus 3.70% in the control group, *p* = 0.0356). A noticeably lower share of HLA-A*03 allele group was also noteworthy (20.0% vs. 27.47% in the control), although it was not statistically significant.

The comparison of patients with illness due to different viral genetic variants showed that HLA-A*03 group alleles were much less common in group 2 convalescents (*p* = 0.0453). This allows us to consider the lower frequency of HLA-A*03 group alleles in those with illness caused by delta VOC strains to be non-random. No differences were found for other allele groups.

The obtained results make it possible to assert that: the HLA-A*02 allele group is as-sociated with a reduced risk of developing illness caused by non-VOC genetic variants; and HLA-A*03 reduces the risk of infection with delta (VOC) genetic variants. At the same time, it is possible that the specificity of HLA-A*30 is associated with a high risk of illness caused by delta variants.

As seen from the data presented in Table 5 regarding the HLA-B locus, no significant differences were found when comparing the control group with patients with infection caused by non-VOC genetic variants. However, group 2 patients (illness caused by the delta VOC) differed significantly from the group of healthy individuals: their HLA-B*57 allele group frequency was more than twice as high, amounting to 13.75% versus 5.43% (*p* = 0.0055). Additionally, their HLA-B*49 allele group frequency was more than twice as high (6.25% versus 2.34%, *p* = 0.0483). The comparison of patient group 1 and group 2 (groups with illness caused by different genetic variants) showed significant differences for HLA-B*49: 0.87% in group 1 versus 6.25% in group 2 (*p* = 0.0136).

When studying the distribution of HLA-DRB1* locus alleles, no differences were found between the control group and either patient group (Table 6). Differences in HLA-DRB1*10 group alleles should not be considered significant due to their rare occurrence among the control group (1.78%).

## 4. Discussion

Analysis of genetic biomarkers, including HLA, is the focus of many researchers seeking to identify possible genetic features that influence the onset, severity, or progression of illness. Works devoted to HLA studies in groups of patients from various European countries have established both predisposing and protective immunogenetic factors, yet the data are at times contradictory. One study examined 619 healthy and 182 patients with SARS-CoV-2 among residents of the island of Sardinia. It was found that the haplotypes HLA-A*02:05, B*58:01, C*07:01, and DRB1*03:01 had a protective effect against SARS-CoV-2 infection in the examined population. Presence of the HLA-DRB1*08:01 allele was an immunogenetic factor that had a negative impact on disease course [10].

Another study was performed in northern Italy [11] when examining 1017 healthy individuals and 99 patients with SARS-CoV-2. A significant association between the disease development and the HLA alleles DRB1*15:01, DQB1*06:02, and B*27:07 was found. Another group of authors did not establish any association between the genotype HLA-B27* and COVID-19 incidence and/or severity in patients with spondyloarthritis [12].

Researchers in Iceland analyzed SARS-CoV-2 reactive T-cell responses in 768 convalescent SARS-CoV-2-infected (cases) and 500 uninfected (controls) Icelanders. The strongest association was between increasing IFN-γ secreting N-reactive CD8^+^ T-cell responses and HLA-B*07:02, followed by HLA-C*07:02 [13].

In 157 adult Caucasian COVID-19 convalescent patients, univariate HLA analysis revealed putatively protective HLA alleles (HLA class II DRB1*01:01 and HLA class I B*35:01, with a pattern of increased DRB1*03:01). They were associated with reduced duration of illness and a greater ability to neutralize the virus compared to non-carriers. Analyses also identified HLA alleles (HLA class II DQB1*03:02, HLA class I B*15:01) not associated with this benefit in this study’s patient cohort [14]. However, the results of such studies often differ; this is associated with population variations in the major histocompatibility complex as well as the sizes of COVID-19 patient cohorts under consideration [15,16,17,18].

The appearance of mutations in the SARS-CoV-2 coronavirus leads to the escape of new strains from CD8+ T-cell responses. Decreased presentation of processed peptides of mutant viral variants by class I HLA molecules leads to: decreased cellular proliferation; decreased IFN-γ production; and decreased cytotoxic activity of CD8+ T cells. This has was shown in cells isolated from patients with COVID-19 with typing for major histocompatibility complex genes [19].

In our previous work [1], which presented the results of a survey of 138 people with SARS-CoV-2 infection, it was shown that: the HLA-A*02 and HLA-A*26 allele groups are protective immunogenetic factors against the development of viral illness among residents of northwest Russia; and the HLA-A*29 allele group is a predisposing immunogenetic factor in relation to viral illness. The current work, performed using a larger number of observations, confirmed the protective properties of HLA-A*02 but only in relation to illness caused by non-VOC genetic variants. Regarding the protective effect of HLA-A*26 in the expanded set of examined patients, only a pattern of lower frequency of this allele group in COVID-19 convalescents with non-VOC genetic variants remained. The A*03 HLA allele group was established as a protective factor against infection caused by delta SARS-CoV-2 strains.

In a study of the association between specific HLA alleles and SARS-CoV-2 susceptibility in a population of 115 United Arab Emirates (UAE) citizens with mild, moderate, or severe SARS-CoV-2 infection, a statistically significant association was observed between the following HLA alleles/supertypes and infection severity. The HLA-B*51:01 and HLA-A*26:01 alleles were significantly more common in non-hospitalized patients, indicating a protective association. Two genotypes (HLA-A*03:01, HLA-DRB1*15:01) and one supertype (B44) showed a significant correlation with disease severity [20].

Those results match ours regarding the protective effect of HLA-A*26 in terms of development of illness caused by SARS-CoV-2 among residents of northwest Russia. In our study, the HLA-A*03 allele group was established as protective against infection with the delta SARS-CoV-2 strain.

Regarding the HLA-A*29 allele group, which was associated with illness risk in our previous study of 138 convalescents, its frequency in the larger group of patients examined here differed from the control group by almost 2-fold: 4.33% compared with 2.47%, although these differences were not statistically significant due to the small number of such patients. Nevertheless, in our opinion, this allows us to reasonably assume that the HLA-A*29 allele group is a predisposing immunogenetic factor in relation to illness caused by non-VOC genetic variants.

Regarding illness caused by delta viral strains, the group of predisposing alleles is larger: HLA-A*30, HLA-B*49, and HLA-B*57. Perhaps this is the reason for the highly contagious nature of this strain. It is interesting to note that one of the predisposing factors, HLA-B*57, is also associated with Behcet’s disease, a systemic vasculitis of unknown etiology, characterized by damage to vessels of any size or type [21]. The results of this study confirm the possible use of HLA typing to differentiate patients who require different strategies for clinical management or, in the future, vaccination.

The study was carried out on the basis of an analysis of the results of the examination of individuals who are potential donors of hematopoietic stem cells and live in St. Petersburg, among whom the vast majority (95.6%) are of Russian nationality. As such, a limitation of the study is generalizability of the findings to other populations.

## 5. Conclusions

Among residents of northwest Russia, HLA-A*02 is only associated with protection against infection caused by non-VOC SARS-CoV-2 genetic variants but is not protective against delta viral strains;HLA-A*03 is associated with a reduced risk of infection by delta SARS-CoV-2 strains;The HLA-A*30, HLA-B*49, and HLA-B*57 allele groups are predisposing factors for infection by delta (B.1.617.2) viral strains.

## Figures and Tables

**Figure 1 ijerph-19-14024-f001:**
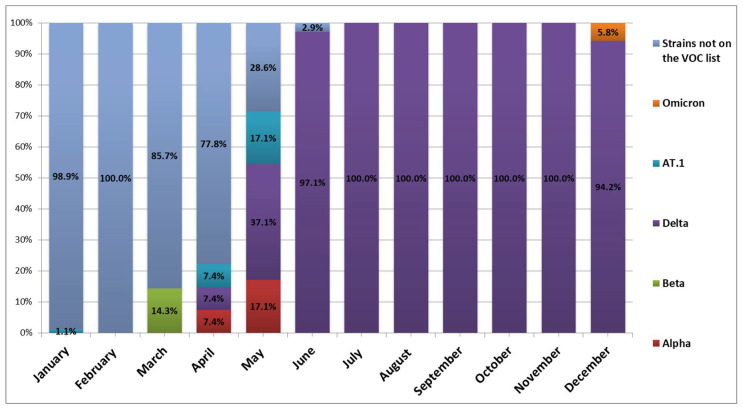
Monitoring dynamics of SARS-CoV-2 genetic variants circulating in St. Petersburg in 2021.

**Table 1 ijerph-19-14024-t001:** Frequency of HLA-A* allele groups among the overall COVID-19 convalescent group (*n* = 484) compared with controls (*n* = 1456).

HLA-A* Allele Group	Frequency in the Overall Convalescent Group, *n* (%)	Frequency in Controls, *n* (%)	*p*
HLA-A*01	122 (25.29)	314 (21.57)	0.1022
HLA-A*02	220 (45.52)	753 (51.72)	0.0182
HLA-A*03	151 (31.26)	400 (27.47)	0.1166
HLA-A*11	58 (11.95)	152 (10.44)	0.3532
HLA-A*23	16 (3.22)	61 (4.19)	0.4233
HLA-A*24	111 (22.99)	289 (19.85)	0.1537
HLA-A*25	36 (7.36)	138 (9.48)	0.1985
HLA-A*26	33 (6.90)	132 (9.07)	0.1329
HLA-A*29	16 (3.22)	36 (2.47)	0.3311
HLA-A*30	22 (4.60)	54 (3.71)	0.4180
HLA-A*31	18 (3.68)	50 (3.43)	0.7758
HLA-A*32	27 (5.52)	56 (3.85)	0.1189
HLA-A*33	16 (3.22)	52 (3.57)	0.8868
HLA-A*36	0 (0.00)	1 (0.07)	1
HLA-A*66	9 (1.84)	18 (1.24)	0.3687
HLA-A*68	33 (6.90)	108 (7.42)	0.7617
HLA-A*69	0 (0.00)	3 (0.21)	1

**Table 2 ijerph-19-14024-t002:** Frequency of HLA-B* allele groups among the overall COVID-19 convalescent group (*n* = 484) compared with controls (*n* = 1456).

HLA-B* Allele Group	Frequency in the Overall Convalescent Group *n* (%)	Frequency in Controls *n* (%)	*p*
HLA-B*07	124 (25.69)	377 (25.89)	0.9522
HLA-B*08	50 (10.32)	181 (12.43)	0.2254
HLA-B*13	67 (10.78)	175 (12.02)	0.3022
HLA-B*14	22 (4.59)	57 (3.91)	0.5952
HLA-B*15	75 (14.45)	179 (12.29)	0.0738
HLA-B*18	60 (12.39)	199 (13.67)	0.5371
HLA-B*27	65 (13.53)	147 (10.10)	0.0438
HLA-B*35	111 (22.94)	304 (20.88)	0.3379
HLA-B*37	10 (2.06)	28 (1.92)	0.8503
HLA-B*38	33 (6.88)	105 (7.21)	0.8385
HLA-B*39	26 (5.28)	70 (4.81)	0.6289
HLA-B*40	50 (10.32)	167 (11.47)	0.5600
HLA-B*41	19 (3.90)	81 (5.56)	0.1913
HLA-B*42	1 (0.23)	3 (0.21)	1
HLA-B*44	92 (19.04)	270 (18.54)	0.8390
HLA-B*45	2 (0.46)	5 (0.34)	0.6877
HLA-B*46	1 (0.23)	3 (0.21)	1
HLA-B*47	2 (0.46)	4 (0.27)	0.6432
HLA-B*48	1 (0.23)	14 (0.96)	0.1353
HLA-B*49	10 (2.06)	34 (2.34)	0.8606
HLA-B*50	10 (2.06)	19 (1.30)	0.2776
HLA-B*51	45 (9.40)	134 (9.20)	0.9280
HLA-B*52	17 (3.44)	59 (4.05)	0.6856
HLA-B*53	0 (0.00)	1 (0.07)	1
HLA-B*54	1 (0.23)	2 (0.14)	0.5774
HLA-B*55	6 (1.15)	38 (2.61)	0.1105
HLA-B*56	16 (3.21)	34 (2.34)	0.2480
HLA-B*57	36 (7.34)	79 (5.43)	0.1192
HLA-B*58	13 (2.75)	30 (2.06)	0.4752
HLA-B*59	0 (0.00)	1 (0.07)	1
HLA-B*67	0 (0.00)	1 (0.07)	1
HLA-B*73	0 (0.00)	2 (0.14)	1
HLA-B*81	0 (0.00)	1 (0.07)	1

**Table 3 ijerph-19-14024-t003:** Frequency of HLA-DRB1* allele groups among the overall COVID-19 convalescent group (*n* = 484) compared with controls (*n* = 1456).

HLA-DRB1* Allele Group	Frequency in the Overall Convalescent Group *n* (%)	Frequency in Controls *n* (%)	*p*
HLA-DRB1*01	125 (25.92)	345 (23.70)	0.3584
HLA-DRB1*03	70 (14.45)	237 (16.28)	0.3883
HLA-DRB1*04	85 (17.66)	297 (20.40)	0.1871
HLA-DRB1*07	133 (27.52)	374 (25.69)	0.4381
HLA-DRB1*08	39 (8.03)	86 (5.91)	0.1084
HLA-DRB1*09	11 (2.29)	36 (2.47)	1
HLA-DRB1*10	0 (0.00)	26 (1.79)	0.0008
HLA-DRB1*11	112 (23.17)	334 (22.94)	0.9502
HLA-DRB1*12	22 (4.59)	56 (3.85)	0.5050
HLA-DRB1*13	114 (23.62)	364 (25.00)	0.5430
HLA-DRB1*14	12 (2.52)	46 (3.16)	0.5384
HLA-DRB1*15	135 (27.98)	413 (28.37)	0.8613
HLA-DRB1*16	41 (8.49)	126 (8.65)	1

**Table 4 ijerph-19-14024-t004:** Frequency of HLA-A* allele groups in the 1st (*n* = 231) and 2nd (*n* = 80) COVID-19 convalescent groups compared with controls (*n* = 1456).

HLA-A* Allele Group	Frequency in Group 1 *n* (%)	Frequency in Group 2 *n* (%)	Frequency in Controls (3) *n* (%)	*p* (1–3)	*p* (2–3)	*p* (1–2)
HLA-A*01	60 (26.00)	23 (28.75)	314 (21.57)	0.1470	0.1292	0.6609
HLA-A*02	98 (42.42)	41 (51.25)	753 (51.72)	0.0088	1	0.1927
HLA-A*03	75 (32.47)	16 (20.0)	400 (27.47)	0.1344	0.1565	0.0453
HLA-A*11	30 (13.00)	9 (11.25)	152 (10.44)	0.2536	0.8506	0.8450
HLA-A*23	5 (2.16)	1 (1.25)	61 (4.19)	0.1982	0.3719	1
HLA-A*24	47 (20.35)	21 (26.25)	289 (19.85)	0.8594	0.1965	0.2755
HLA-A*25	20 (8.66)	8 (10.00)	138 (9.48)	0.8079	0.8446	0.8208
HLA-A*26	16 (6.93)	6 (7.50)	132 (9.07)	0.3185	0.8404	0.8052
HLA-A*29	10 (4.33)	2 (2.50)	36 (2.47)	0.1246	1	0.7373
HLA-A*30	10 (4.33)	7 (8.75)	54 (3.71)	0.5820	0.0356	0.1547
HLA-A*31	6 (2.60)	2 (2.50)	50 (3.43)	0.6918	1	1
HLA-A*32	15 (6.50)	3 (3.75)	56 (3.85)	0.0758	1	0.5783
HLA-A*33	8 (3.46)	4 (5.00)	52 (3.57)	1	0.5314	0.5130
HLA-A*36	0 (0.00)	0 (0.00)	1 (0.07)	1	1	-
HLA-A*66	6 (2.60)	0 (0.00)	108 (7.42)	0.1269	0.6200	0.3443
HLA-A*68	20 (8.66)	5 (6.25)	3 (0.21)	0.5039	0.8286	0.6357
HLA-A*69	0 (0.00)	0 (0.00)	314 (21.57)	1	1	-

**Table 5 ijerph-19-14024-t005:** Frequency of HLA-B* allele groups in the 1st (*n* = 231) and 2nd (*n* = 80) COVID-19 convalescent groups compared with controls (*n* = 1456).

HLA-B* Allele Group	Frequency in Group 1 *n* (%)	Frequency in Group 2 *n* (%)	Frequency in Controls (3) *n* (%)	*p* (1–3)	*p* (2–3)	*p* (1–2)
HLA-B*07	61 (26.41)	19 (23.75)	377 (25.89)	0.8718	0.7930	0.7668
HLA-B*08	25 (10.82)	13 (16.25)	181 (12.43)	0.5884	0.3014	0.2344
HLA-B*13	19 (8.23)	11 (13.75)	175 (12.02)	0.0967	0.5992	0.1859
HLA-B*14	10 (4.33)	2 (2.50)	57 (3.91)	0.7180	0.7658	0.7373
HLA-B*15	30 (12.99)	11 (13.75)	179 (12.29)	0.7476	0.7264	0.8495
HLA-B*18	30 (12.99)	11 (13.75)	199 (13.67)	0.8367	1	0.8495
HLA-B*27	23 (9.96)	9 (11.25)	147 (10.10)	1	0.7040	0.8310
HLA-B*35	55 (23.81)	13 (16.25)	304 (20.88)	0.3410	0.3945	0.2088
HLA-B*37	7 (3.03)	3 (3.75)	28 (1.92)	0.3148	0.2164	0.7211
HLA-B*38	20 (8.66)	5 (6.25)	105 (7.21)	0.4186	1	0.6357
HLA-B*39	12 (5.19)	2 (2.50)	70 (4.81)	0.7434	0.5821	0.5314
HLA-B*40	21 (9.09)	8 (10.00)	167 (11.47)	0.3128	0.8565	0.8247
HLA-B*41	8 (3.46)	2 (2.50)	81 (5.56)	0.2077	0.3147	1
HLA-B*42	1 (0.43)	0 (0.00)	3 (0.21)	0.4455	1	1
HLA-B*44	43 (18.61)	13 (16.25)	270 (18.54)	1	0.7668	0.7366
HLA-B*45	0 (0.00)	1 (1.25)	5 (0.34)	1	0.2749	0.2572
HLA-B*46	2 (0.87)	0 (0.00)	3 (0.21)	0.1410	1	1
HLA-B*47	2 (0.87)	0 (0.00)	4 (0.27)	0.1930	1	1
HLA-B*48	0 (0.00)	0 (0.00)	14 (0.96)	0.2390	1	-
HLA-B*49	2 (0.87)	5 (6.25)	34 (2.34)	0.2178	0.0483	0.0136
HLA-B*50	5 (2.16)	1 (1.25)	19 (1.30)	0.3619	1	1
HLA-B*51	17 (7.36)	10 (12.5)	134 (9.20)	0.4560	0.3231	0.1703
HLA-B*52	7 (3.03)	2 (2.50)	59 (4.05)	0.5840	0.7671	1
HLA-B*53	0 (0.00)	0 (0.00)	1 (0.07)	1	1	-
HLA-B*54	1 (0.43)	0 (0.00)	2 (0.14)	0.3573	1	1
HLA-B*55	2 (0.87)	0 (0.00)	38 (2.61)	0.1577	0.2594	1
HLA-B*56	9 (3.90)	1 (1.25)	34 (2.34)	0.1749	1	0.4621
HLA-B*57	18 (7.79)	11 (13.75)	79 (5.43)	0.1693	*	0.1221
HLA-B*58	6 (2.60)	0 (0.00)	30 (2.06)	0.6217	0.4004	0.3443
HLA-B*59	0 (0.00)	0 (0.00)	1 (0.07)	1	1	-
HLA-B*67	0 (0.00)	0 (0.00)	1 (0.07)	1	1	-
HLA-B*73	0 (0.00)	0 (0.00)	2 (0.14)	1	1	-
HLA-B*81	0 (0.00)	0 (0.00)	1 (0.07)	1	1	-

**Table 6 ijerph-19-14024-t006:** Frequency of HLA-DRB1* allele groups in the 1st (*n* = 231) and 2nd (*n* = 80) COVID-19 convalescent groups compared with controls (*n* = 1456).

HLA-DRB1* Allele Group	Frequency in Group 1 *n* (%)	Frequency in Group 2 *n* (%)	Frequency in Controls (3) *n* (%)	*p* (1–3)	*p* (2–3)	*p* (1–2)
HLA-DRB1*01	52 (22.51)	15 (18.75)	345 (23.70)	0.7388	0.3451	0.5310
HLA-DRB1*03	41 (17.75)	16 (20.00)	237 (16.28)	0.5675	0.3564	0.7375
HLA-DRB1*04	47 (20.35)	16 (20.00)	297 (20.40)	1	1	1
HLA-DRB1*07	65 (28.14)	23 (28.75)	374 (25.69)	0.4209	0.5152	1
HLA-DRB1*08	15 (6.49)	4 (5.00)	86 (5.91)	0.7648	1	0.7897
HLA-DRB1*09	5 (2.16)	2 (2.50)	36 (2.47)	1	1	1
HLA-DRB1*10	0 (0.00)	1 (1.25)	26 (1.79)	0.0397	1	0.2572
HLA-DRB1*11	46 (19.91)	17 (21.25)	334 (22.94)	0.3510	0.7860	0.8719
HLA-DRB1*12	7 (3.03)	3 (3.75)	56 (3.85)	0.7084	1	0.7211
HLA-DRB1*13	56 (24.24)	16 (20.00)	364 (25.00)	0.8700	0.3534	0.5387
HLA-DRB1*14	9 (3.90)	4 (5.00)	46 (3.16)	0.5492	0.3268	0.7465
HLA-DRB1*15	66 (28.57)	26 (32.5)	413 (28.37)	0.9376	0.4462	0.5700
HLA-DRB1*16	22 (9.52)	6 (7.50)	126 (8.65)	0.6187	0.8400	0.6580

## Data Availability

Not applicable.

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
