# Peer review of "HLA Genotypes in Patients with Infection Caused by Different Strains of SARS-CoV-2"

_ijerph, 2022, doi:10.3390/ijerph192114024_

Round 1

Reviewer 1 Report

The manuscript by Bubnova et al. investigated HLA polymorphism in northwest Russia and compared allele frequency between delta variant (VOC), non-VOC, and control groups. The manuscript addresses an important question, particularly for the local region, and will be of interest to the journal readers. However, I have a few concerns regarding the manuscript. 

  1. The language of the manuscript could be improved. 
  2. The significance of the study needs to be strengthened in the introduction. 
  3. To improve readability, the authors can consider splitting the Materials and Methods section and Result Section into subsections with subtitles, using 2.1, 2.2, etc. 
  4. How the allele frequency was calculated from the Method section was unclear. Moreover, please provide a brief description of the correction for the small sample size, as it was only mentioned and cited. Should the difference value be adjusted for age and sex? 
  5. Line 125, should it be 'significant at X2 >= 3.84 (p<0.05)'?
  6. It makes more sense to move Figure 1 to the Result section. Add a table of demographics for all study groups (lines 86-92) and move this paragraph to Result.   
  7. For ease of understanding, please use a statistical computing package or software to convert the X2 values to p values in all tables. 
  8. Please include a paragraph about the study's limitations at the end of the Discussion. 

Minor observations

  1. line 62, two period

Author Response

Please seethe attachment

Reviewer 2 Report

1. The study is interesting, paper is well written and well designed.

2. The number of samples is high and that is very good for statistics and its relevance. It is very nice to see that the results are corroborating with other studies done in different populations. This provides a strong basis for the association.

3. However, there are many studies that have found different alleles that provide protection and susceptibility, which is expected and that accounts for the population differences. It will be good to see some more discussion of these studies in the discussion section. It will add to the value of this study.

For example: PMID 35395388: This group found HLA A 68 to be protective in the Mexican population.  You may include others that may be relevant as well. 

4. The authors have described and presented the data well. Just a few things can be improved for clarity sake. 

Table 1 alignment has issues. 

Description of the tables, legends and asterisks can be improved.

It is not clear in the description of the Tables, whether the calculated frequency is with the N of covid group and control group individually or out of total population?

Please give the x2 significance in the legend as well. 

Line 125 states: "Differences were regarded as significant at χ2 ≤ 3.84 (P<0.05)." 

However, the values that are marked significant are x2 >= 3.84

Perhaps it will be better to use a different symbol than the asterisk to mark those values that are significant x2 values, as the asterisk is part of the HLA typing nomenclature.

In addition, those values that are statistically significant, but not to be considered significant as described in text, can be marked differently for e.g HLA DRB1 10. 
